# Anthropometrics, Dietary Intake and Body Composition in Urea Cycle Disorders and Branched Chain Organic Acidemias: A Case Study of 18 Adults on Low-Protein Diets

**DOI:** 10.3390/nu14030467

**Published:** 2022-01-21

**Authors:** Giorgia Gugelmo, Livia Lenzini, Francesco Francini-Pesenti, Ilaria Fasan, Paolo Spinella, Romina Valentini, Angela Miraval, Angelo Avogaro, Nicola Vitturi

**Affiliations:** 1Division of Clinical Nutrition, Department of Medicine-DIMED, University Hospital, University of Padova, 35128 Padova, Italy; giorgia.gugelmo@aopd.veneto.it (G.G.); francescofrancini@yahoo.it (F.F.-P.); ilaria.fasan@aopd.veneto.it (I.F.); paolo.spinella@unipd.it (P.S.); romina.valentini@unipd.it (R.V.); angelamirav@gmail.com (A.M.); 2Department of Medicine-DIMED, University Hospital, University of Padova, 35128 Padova, Italy; livia.lenzini@unipd.it; 3Division of Metabolic Diseases, Department of Medicine-DIMED, University Hospital, University of Padova, 35128 Padova, Italy; angelo.avogaro@unipd.it

**Keywords:** low-protein diet, nutritional status, adult, inherited metabolic disorders

## Abstract

Low-protein diets (LPDs) are the mainstream treatment for inborn errors of intermediary protein metabolism (IEIPM), but dietary management differs worldwide. Most studies have investigated pediatric populations and their goals such as growth and metabolic balance, showing a tendency toward increasing overweight and obesity. Only a few studies have examined nutritional status and dietary intake of adult IEIPM patients on LPDs. We assessed nutritional parameters (dietary intake using a 7-day food diary record, body composition by bioimpedance analysis, and biochemical serum values) in a group of 18 adult patients with urea cycle disorders (UCDs) and branched chain organic acidemia (BCOA). Mean total protein intake was 0.61 ± 0.2 g/kg/day (73.5% of WHO Safe Levels) and mean natural protein (PN) intake was 0.54 ± 0.2 g/kg/day; 33.3% of patients consumed amino acid (AA) supplements. A totally of 39% of individuals presented a body mass index (BMI) > 25 kg/m^2^ and patients on AA supplements had a mean BMI indicative of overweight. All patients reported low physical activity levels. Total energy intake was 24.2 ± 5 kcal/kg/day, representing 72.1% of mean total energy expenditure estimated by predictive formulas. The protein energy ratio (P:E) was, on average, 2.22 g/100 kcal/day. Plasmatic levels of albumin, amino acids, and lipid profiles exhibited normal ranges. Phase angle (PA) was, on average, 6.0° ± 0.9°. Fat mass percentage (FM%) was 22% ± 9% in men and 36% ± 4% in women. FM% was inversely and significantly related to total and natural protein intake. Data from IEIPM adults on LPDs confirmed the pediatric trend of increasing overweight and obesity despite a low energy intake. A low protein intake may contribute to an increased fat mass. Nutritional parameters and a healthy lifestyle should be routinely assessed in order to optimize nutritional status and possibly reduce risk of cardiovascular degenerative diseases in adult UCD and BCOA patients on LPDs.

## 1. Introduction

Low-protein diets (LPDs) are the main treatment for inborn errors of intermediary protein metabolism (IEIPM), such as urea cycle disorders (UCDs) and branched chain organic acidemia (BCOA).

There is a consensus on limiting the natural protein intake, both in pediatric and adult populations, while the use of amino acid formulas in UCDs and BCOA is still discussed. LPDs are individualized for each patient, considering clinical status and individual tolerance to toxic metabolites [1]. LPDs may potentially expose patients to the risk of deficiencies in essential micronutrients, amino acids, and fatty acids [1].

Recent guidelines [2,3] for the management of patients with UCDs and BCOA have advised a low-protein diet and referred to the World Health Organization (WHO), Food and Agriculture Organization of the United Nations (FAO), and United Nations University (UNU) protein and energy requirements [4] (in adult populations, protein requirement is 0.83 g/kg/day; energy requirement is related to physical activity level and body weight). The use of amino acid formulas is advised when protein requirements are not reached with only natural protein, calculated as the provision of 20–30% of the total protein intake. The amount of these formulas is still debated among different metabolic centers [2,3,5].

Another group of IEIPM treated with LPDs is aminoacidopathies such as phenylketonuria (PKU), tyrosinemia type 1 (TYR-1), homocystinuria (HCU), and maple syrup urine diseases (MSUDs); in these disorders, LPD guidelines are characterized by a very low natural protein tolerance and a clear prescription of amino acid (AA) formulas specific to each disease [6,7,8,9].

Some studies of dietetic treatment in IEIPM have investigated pediatric populations, aiming to evaluate growth and metabolic balance [10,11]; in particular, in pediatric and 20-year-old adult methylmalonic and propionic acidemia patients, a higher protein prescription was correlated with more acute metabolic decompensations, mitochondrial complications, and lower height and cognition [12]. A recent meta-analysis of aminoacidopathies showed a significantly increased BMI in classical PKU pediatric patients compared to healthy controls [13].

During chronic management of IEIPM, protein adequacy should be carefully monitored, and clinical status and biochemical markers (prealbumin, albumin, and plasmatic amino acids) should be measured routinely [1]. IEIPM clinicians and dietitians should be vigilant toward appropriate weight gain during pediatric age and the risk of long-term overweight and obesity [1].

In European metabolic centers, the implementation of newborn screening has increased the number of treated patients and their life expectancy [14], raising the issue of evaluating long-term complications in adults.

Nutritional outcomes in adult UCD and BCOA patients treated with LPDs have not been systematically studied, and there is a lack of studies investigating physical activity levels as well as overweight or obesity status and cardiovascular risk parameters [15].

Data from adults with PKU on LPDs showed a high prevalence of obesity and overweight risk, in particular in women [15,16,17].

Since growth outcomes in pediatric UCD and BCOA patients are not ideal even if protein and energy intake meet recommendations [10], it is necessary to better comprehend how nutritional parameters in adult patients can influence the risk of obesity and cardiovascular diseases. Little is known about the long-term effects of LPDs on nutritional status in IEIPM adult patients, especially in comparison to healthy individuals.

Given that data from the literature are still lacking in adult patients, we performed a retrospective study to assess nutritional status in a cohort of adults affected by UCDs and BCOA. We assessed anthropometrics, dietary intake, and body composition, and we investigated the relationship of these parameters to nutritional outcomes to better understand the possible long-term nutritional implications of LPDs.

## 2. Materials and Methods

We retrospectively assessed data from adult patients affected by UCD and BCOA who were followed at the Inherited Metabolic Rare Diseases Adult Centre of the University Hospital of Padova.

No exclusion criteria were applied.

The complete medical history and the physical assessment were recorded from each patient.

The nutritional assessment included a 7 day food diary record, in which the amount of every meal’s food component was expressed in grams. These data were then analyzed by Metadieta^®^ software (Meteda—METEDA S.r.l.Via S. Pellico, 4 63074 San Benedetto del Tronto (AP),Italy) to calculate the average dietary intake over the 7-day period. The Dietetic Reference Values (DRVs) of the European Food Safety Authority (EFSA) were considered for requirements of macro- and micronutrients [18]. Energy requirements were estimated by using Harris and Benedict’s predictive formula for resting energy expenditure (REE), then increased by physical activity level (PAL). For single amino acid requirements, we referred to WHO reports (mg/kg/day) [4].

A mechanical scale with movable weights and an altimeter by Seca^®^ (weight precision: 50 g; height precision 0.5 cm) were used for the weight and height assessments, respectively.

Body composition was evaluated by measurement of impedance parameters (resistance and reactance) by an Akern^®^ BIA 101 New Edition (sinusoidal 50 kHz waveform current, intensity 0.8 A). Bioimpedance measurement was assessed after fasting. Qualitative and quantitative body composition was elaborated by BODYGRAM™ and compared with normal Caucasian population values [19], in particular for fat mass (FM), free fat mass (FFM), and phase angle (PA), and derived data such as the free fat mass index (FFMI) and the fat mass index (FMI) [20].

Blood tests were collected at the same time as clinical and nutritional assessments by venous puncture after fasting for at least 12 h. The serum parameters were determined as follows: albumin, transthyretin, total protein, amino acid profile, transaminases, glucose, triglycerides (TG), total cholesterol (TC), HDL cholesterol (HDL-C), and LDL cholesterol (LDL-C)).

Data analyses were performed using Microsoft^®^ Excel 2019 and Prism 9. A descriptive statistical study of the sample was completed by using the parameters of centralization (mean and median) and dispersion (standard deviation, maximum, and minimum), according to variable type.

T-tests were used to compare means of different subgroups, and Pearson’s test was used to establish correlations between FM% and total protein intake, natural protein intake, and BMI (*p* value < 0.05, confidence interval 0.95).

## 3. Results

We recorded data from 18 adult patients with UCDs and BCOA.

### 3.1. General Characteristics, Dietray Intake, Biochemical Parameters and Body Composition

#### 3.1.1. Subject Characteristics

The general characteristics of the subjects are summarized in Table 1.

All individuals were Caucasian on LPDs; they each had a pediatric diagnosis based on clinical symptoms, and none were detected using NBS. A total of 33.3% of patients were affected by organic acidemias and 66.7% by UCDs. Their median age was 28.6 ± 8 years. Pharmacological and nutritional therapy started between 0 and 7 years of age. All patients were fed orally. None of the individuals experienced acute metabolic decompensation during the two years preceding the evaluation. All patients reported low physical activity level (PAL) and hypokinetic lifestyle; one subject presented difficulty in ambulation.

The median BMI was 24.9 ± 3.8 kg/m^2^; 55.6% of individuals presented normal BMI (18.5 < BMI < 24.9 kg/m^2^), 27.7% were overweight (25 < BMI < 29.9 kg/m^2^), 11.1% presented obesity I grade (30 < BMI < 34.9 kg/m^2^), and 5.5% were underweight (BMI < 18.5 kg/m^2^).

#### 3.1.2. Dietary Intake

The LPDs included low-protein foods in 61% of patients; 33.3% consumed amino acid (AA) supplements specific to disease, and 66.7% followed LPDs with only natural protein intake.

All AA formulas used contained additional micronutrients and carbohydrates (maltodextrins).

The mean daily natural protein (NP) intake was 0.54 ± 0.18 g/kg/day; the mean protein equivalent (PE) provided by amino acid supplements was 0.22 ± 0.13 g/kg/day (for those on AA supplements). As reported in Figure 1a, the mean total protein intake was 0.61 ± 0.18 g/kg/day and, compared to WHO Safe Levels for adult subjects (0.83 g/kg/day), it provided 73.5% of recommended values [4].

The group of patients on LPDs with just NP showed an NP intake of 0.58 ± 0.18 g/kg/day (Figure 1b). Patients on LPDs with AA supplements presented an NP intake of 0.44 ±0.1 g/kg/day and a total protein intake of 0.66 ±0.16 g/kg/day, closer to WHO Safe Levels [4] (Figure 1c).

Patients affected by BCOA showed a total protein intake (0.74 ±0.12 g/kg/day) higher than that of UCD patients (0.54 ± 0.16 g/kg/day), given that both group of patients received AA supplementation at the same rate (33.3% of UCD and OA patients). 

Patients on AA supplements presented a mean BMI of 26.1 ± 3.2 kg/m^2^; patients without AA supplements presented a mean BMI of 24.3 ± 4 kg/m^2^.

Total daily energy intake (TDEE) was 24.2 ± 5.4 kcal/kg/day, representing 72.1% of the mean total energy expenditure estimated by reference standards. The group of patients on LPDs with just NP presented a TDEE of 23.2 kcal/kg/day, and the group with AA supplements showed a TDEE of 26.3 kcal/kg/day, without significant differences between the two. 

The protein–energy ratio (P:E) was, on average, 2.22 g/100 kcal/day.

The mean intake of single essential AAs from natural protein food is reported in Figure 2, compared to reference values [4]. 

Mean leucine, isoleucine, and valine intakes were lower than requirements [4], as well as mean intakes for lysine, methionine, and threonine. Mean tryptophan and phenylalanine intakes were closer to reference values. Only 27.8% of patients met branched-chain AA (BCAA) reference values. In the group of patients with just NP (Figure 3a), the BCAA median intake was lower than requirements (leucine intake = 41.1 mg/kg/day, isoleucine intake = 22.4 mg/kg/day, and valine intake = 24.9 mg/kg/day). In the AA supplementation group, instead, the median BCAA intake exceeded the requirements (leucine intake= 67.9 mg/kg/day, isoleucine intake = 32 mg/kg/day, and valine intake = 41.9 mg/kg/day), while median BCAA intake from NP was lower than requirements (Figure 3b). BCAA intake from natural proteins was significantly lower in patients with AA supplementation than in those with only NP intake (*p* = 0.01).

The averages total carbohydrate and fat percentages of energy intake were, respectively, 47.3% (of which sugars were 14.4% and fiber was 14 g/day) and 31.1% (EFSA reference intake (RI) range for total carbohydrates: 45–60% of energy intake; EFSA RI range for total fat: 20–35% of energy intake) [18]. In patients with AA supplementation, the energy distribution was as follows: fat: 27%; carbohydrates: 44.9% (sugars 20.4% and fiber 15 g/day); in patients without supplementation: fat: 33%; carbohydrates: 48.5% (sugars 13%, fiber 13.4 g/day). The sugar percentage of energy intake was significantly higher in patients with AA supplementation (*p* = 0.03).

For those on AA supplements, AA formulas provided, on average, 8% of the total energy intake (140 kcal/day, maltodextrins 20.7 g, lipids 0 g, and protein equivalent 14.2 g).

All patients received micronutrient supplementation from vitamins and mineral supplements and/or AA-specific supplements enriched in micronutrients. The micronutrient supplementation used was specific to patients on LPDs but not to those of adult age. 

#### 3.1.3. Biochemical Parameters

All patients’ biochemical parameters are summarized in Table 2.

Regarding BCAA plasmatic levels, 56% of patients were within reference values for leucine, 72% for isoleucine, and only 39% for valine.

A total of 83% presented phenylalanine and threonine within reference values, and 78% for methionine.

AA plasmatic levels are reported in Table 3. Interestingly, the median BCAA plasmatic levels in patients with AA supplementation were lower than reference values; instead, the median BCAA plasmatic levels of patients without AA supplementation were within reference range values, even though we did not find a statistically significant difference between the two subgroups.

#### 3.1.4. Body Composition

With regard to BIA analysis, the phase angle (PA) was 6.0° ± 0.9° (women: 5.7° ± 0.7°; men: 6.2° ± 0.9°), as reported in Figure 4a with reference values [20].

Fat mass (FM) percentage was 36 ± 4 FM% in women and 22 ± 9 FM% in men, which are higher than reference values [19] (Figure 4b).

The FM index (FMI) was calculated in relation to height squared: 10 ± 2 kg/m^2^ in women and 5.7 ± 3 kg/m^2^ in men (Figure 4c). FM% and FMI were, on average, higher than normal reference values [19], especially in the group of patients with AA supplementation (FM% = 27 ± 10).

The FFM index (FFMI) related to height squared was 17.4 ± 1 kg/m^2^ in women and 19 ± 1 kg/m^2^ in men (Figure 4d).

Correlations between natural and total protein intake (g/kg/day) and FM% are shown in Figure 5a,b: our patients’ trends revealed that the increase in total protein intake corresponded to a decrease in FM% (r = −0.560, *p* = 0.037), as well as to a decrease in natural protein intake (r = −0.599, *p* = 0.024).

A positive correlation between BMI and FM% was found (r = 0.843, *p* < 0.001) (Figure 5c).

## 4. Discussion

Our study evaluated anthropometrics, dietary intake, and body composition in adult patients with UCDs and BCOA on LPDs since a pediatric age. Adults with UCDs and BCOA represent a new growing population due to newborn screening and progress in medicine that have allowed these patients to become adults [14]. Both LPDs and drugs have helped to extend life expectancy, but, until now, few studies have investigated nutritional status and dietary adequacy in these patients, who must continue lifelong treatment with LPDs [21].

In our patients, natural and total protein average intake were lower than WHO Safe Levels, but plasmatic levels of albumin and prealbumin were within the range in all individuals. Energy intake was lower than total daily energy intake (TDEE) estimated by predictive equations in all subjects, both in patients with or without AA supplementation, with no significant difference in median TDEE between the two subgroups. Despite this, the high prevalence of overweight and obesity and no acute metabolic decompensation during periods of observations is reminiscent of a positive energy balance. Moreover, total and natural protein intake were inversely related to FM%, confirming data observed by Evans et al. in a pediatric population [10]. TDEE was calculated by REE from predictive equations multiplied by PAL indicated for standard lifestyles [18]. In our sample, REE from the predictive equation could have been influenced by a low FFMI, affected by low protein intake, and by a PAL lower than standard levels [21]. The TDEE obtained from predictive equations for REE and PAL is further derived from healthy individuals and it may not be a proper reference in these special subjects.

Another factor involved in energy intake being lower than reference values may be an under-reported dietary intake by patients.

Despite AA supplements promoting a higher protein intake, patients treated with AA supplements showed a higher prevalence of overweight or obesity than those fed only with natural protein. Dividing the cohort of patients into two subgroups, with or without AA supplementation, we noted higher BMI, FM%, and plasmatic AA levels (in particular BCAA) in the AA supplementation group, without significant differences in energy intake. Therefore, we questioned what is the best practice in protein prescription, considering not only total protein intake closer to required reference values but also the protein source (natural food vs. AA supplements). As suggested by Francini-Pesenti et al. [21], we hypothesized that, together with a different elemental protein source, a lower natural protein intake can lead to a lower FFMI and a higher FM% in this group by promoting overweight or obesity. A positive energy balance during LPDs can be explained using the protein leverage model that postulates the overconsumption with fats and carbohydrates in response to a reduction in protein intake and vice versa [22].

The higher levels of BMI and FM% observed in patients treated with AA supplements may also be due to different protein sources. Whole protein intake also induces satiety through bioactive peptides derived from intestinal protein digestion [23], which are not produced in the case of AA ingestion.

Another factor possibly influencing the higher BMI in the subgroup with AA supplementation is the energy source of AA supplements, which is often an elemental type of sugar (maltodextrins) as sugar percentage of energy intake in this group of patients was significantly higher.

In the case of MMA/PA patients, overweight could be due to the leucine load of AA supplements, which can represent an anabolic factor promoting increases in BMI and FM% [24], and is also supported by abnormal BCAA ratios in these patients [5]. Recent data from a European Multicentre registry showed that methylmalonic and propionic patients treated with amino acid formulas presented with abnormal plasmatic BCAA ratios, in contrast to the good effects of BCAA-enriched amino acid formulas in UCD patients [5]. In our sample, BCAA plasmatic levels were lower than recommended in many of our patients, especially patients with AA supplementation. In particular, the leucine plasmatic levels in this subgroup were lower than reference values despite a leucine intake (67 mg/kg/day) higher than requirements (48 mg/kg/day). Given the key role of BCAA intake in maintaining a higher FFM, the intake in these patients (both by PN and AA formulas) should be taken into consideration [25]. Moreover, EAA intake and plasmatic levels should be routinely monitored in patients on LPDs, considering the frequent intake of vegetable foods in LPDs, which are good for their low protein content but can result in low biological values and BCAA content.

A P:E ratio of 1.5–2.9 g/100 kcal/day was indicated by Evans et al. [10] for optimal outcomes in a pediatric population, which are correlated to nutritional status and body composition; in adult populations, a clear definition of good outcomes is lacking [26], and therefore, an optimal P:E ratio cannot be identified.

Another difficulty in dealing with adults affected by UCD and BCOA is that WHO Safe Levels for protein intake refer to normal populations with normal levels of physical activity and good representation in FFM, while our patients presented lower PAL and FFM. The role of a lower PAL in altered body composition is also confirmed by recent data from a pediatric population [11]. Whereas reduced protein intake is an essential aspect of metabolic treatment for these patients, the maximum tolerance of protein intake (g/kg) should be tailored to promote FFM together with maintaining good metabolic control. Moreover, as suggested by Rocha et al. for PKU [27], PAL should be routinely assessed and improved in order to obtain a higher FFM. From pediatric data, moderate or vigorous PAL can also lead to higher bone mineral density and FFM [11].

In addition to low PAL and increased BMI, biochemical metabolic parameters such as serum glucose and lipid profiles should be considered to assess degenerative cardiovascular disease risk in the adult population. To assess the prognostic value of overweight in IEIPM patients, its protective effect should also be considered, indicated by the term obesity paradox [28,29]. There are no current studies aimed at evaluating the relationship between BMI and life expectancy in IEIPM adult patients treated with LPDs.

Vitamin and mineral supplements specific to LPDs are needed [2,3,6,7,8,9], but most of these supplements are designed for pediatric ages and are not well-tailored for adulthood.

A strength of this study is that data belong to a single Inherited Metabolic Rare Diseases Adult Centre that follows patients since their transition from a pediatric age and reports the precise intake of consumed (not prescribed) LPDs. An important limitation of this study is the small number of patients examined, due to the low prevalence of adult patients affected by these diseases. Long-term outcomes (i.e., cardiovascular events) could not be examined because our patients were still young (mean age 28 years) and more follow-up is needed. In addition, nutritional compliance with prescribed LPDs and feeding behaviors such as food selectivity and satiety levels were not assessed.

Further studies on nutritional adequacy and nutritional status are needed in adult patients with UCDs and BCOA. We know with certainty that numbers of metabolic patients are growing and their nutritional needs and outcomes are changing over time. Protein intake should meet recommendations [2,3] and AA supplementation should be used when protein needs are not met with natural foods. Periodic assessments of nutritional status, recognition of FFM, intake of macro- versus micronutrients, and risk factors for cardiovascular diseases must be included in routine evaluation in adult centers. Long-term follow-up may give insights into the effects of lifelong LPDs. A new challenge will be better understanding how LPDs can affect sarcopenic processes in elderly UCD and BCOA individuals who have been on LPDs since a pediatric age.

## Figures and Tables

**Figure 1 nutrients-14-00467-f001:**
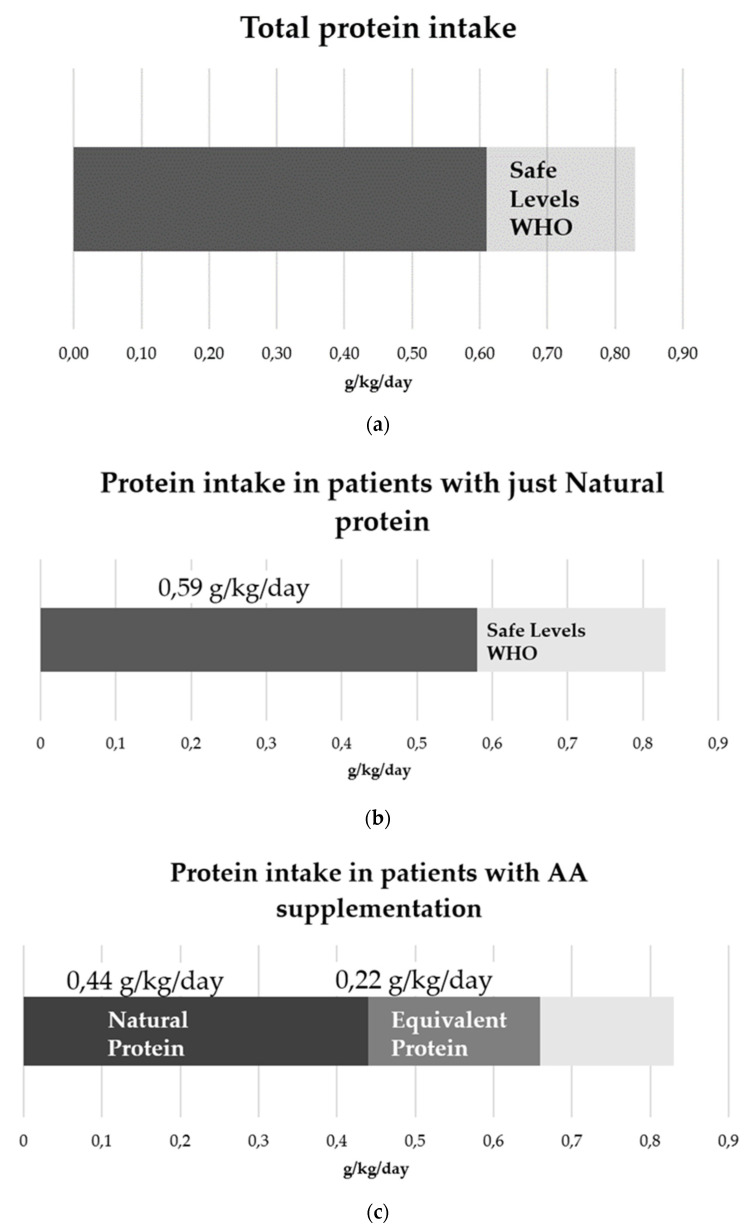
Description of protein intake: (**a**) total protein intake of all subjects (0.61 g/kg/day) and WHO Safe Levels (0.83 g/kg/day); (**b**) protein intake in patients without amino acid supplementation (0.58 g/kg/day) and WHO Safe Levels (0.83 g/kg/day); (**c**) protein intake in patients with amino acid supplementation (natural protein intake = 0.44 g/kg/day, equivalent protein intake = 0.22 g/kg/day, and total protein intake = 0.66 g/kg/day) and WHO Safe Levels (0.83 g/kg/day).

**Figure 2 nutrients-14-00467-f002:**
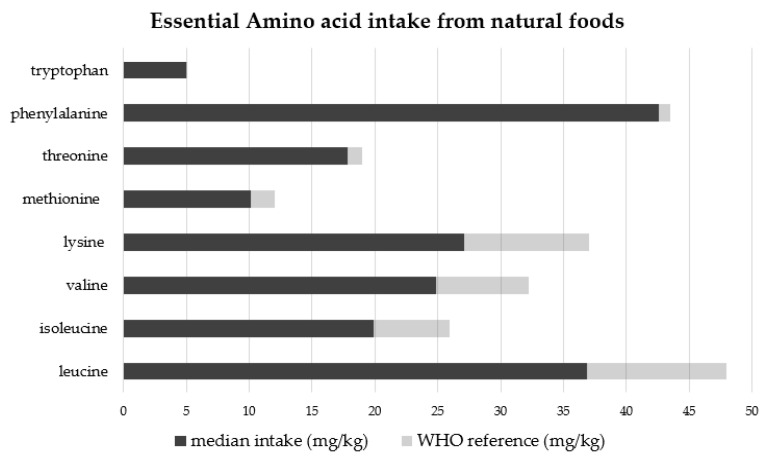
Essential amino acid intakes from natural protein foods compared to reference values (mg/kg/day).

**Figure 3 nutrients-14-00467-f003:**
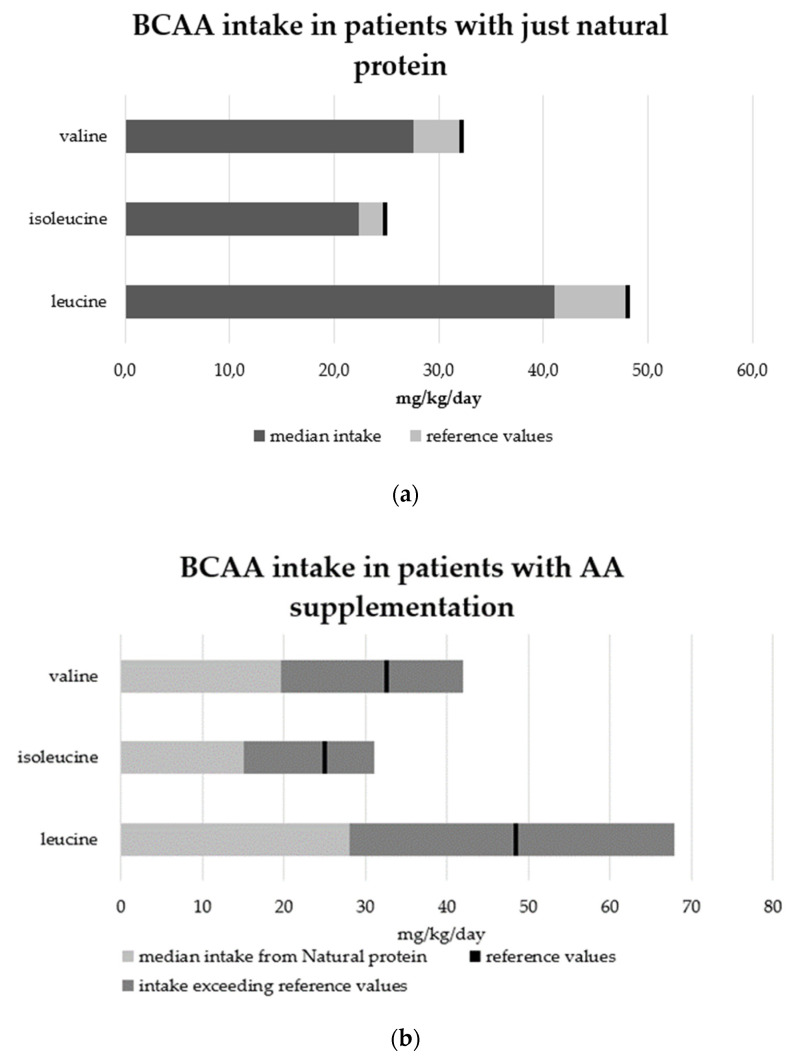
Branched-chain amino acid (BCAA) intakes compared to reference values (mg/kg/day) in a group of patients with just NP intake (**a**) and in a group of patients with AA supplementation (**b**).

**Figure 4 nutrients-14-00467-f004:**
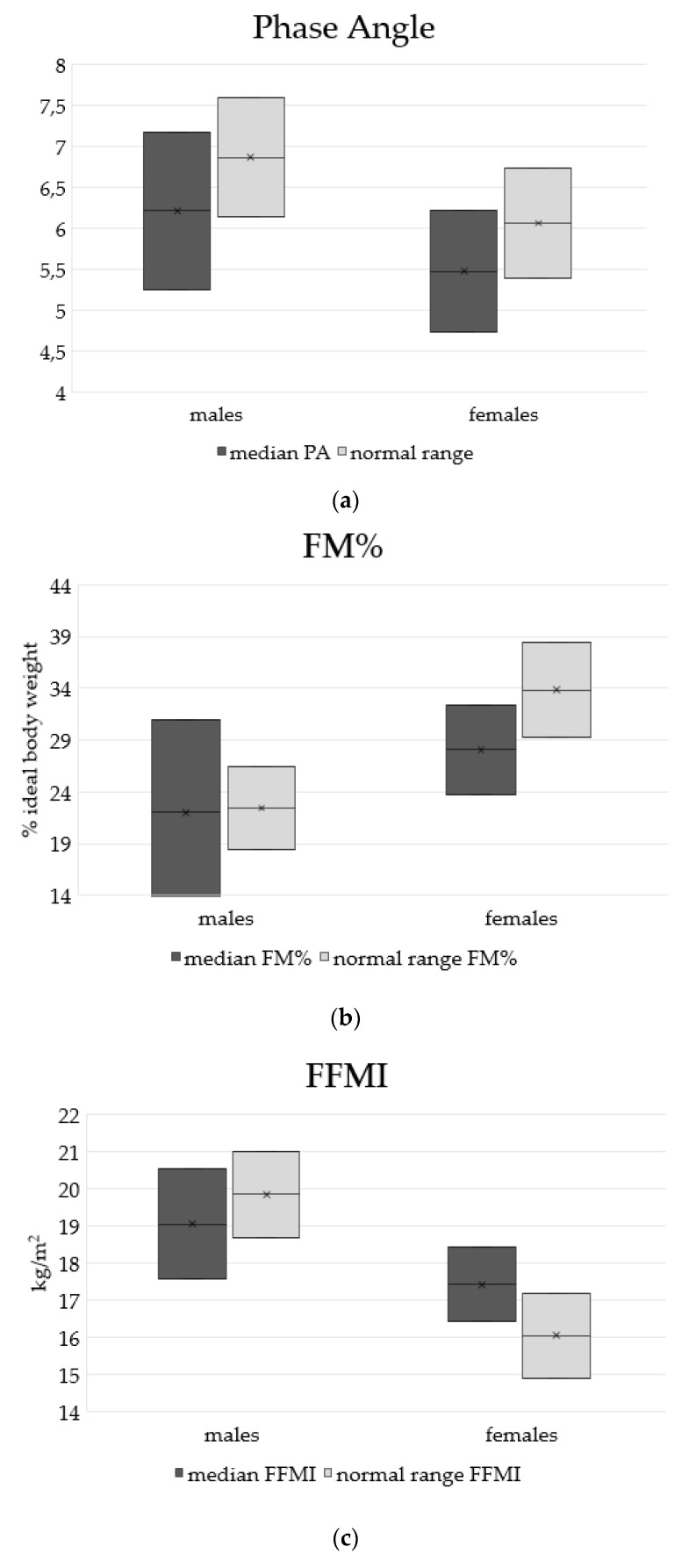
Median patient and reference values for body composition: (**a**) phase angle (men 6.2 ± 1; women 5.5 ± 0.7); (**b**) fat mass percentage (men 22 ± 9; women 36 ± 4); (**c**) free fat mass index (men 19 ± 1.5; women 17 ± 1); (**d**) fat mass index (men 5.7 ± 3; women 10 ± 2).

**Figure 5 nutrients-14-00467-f005:**
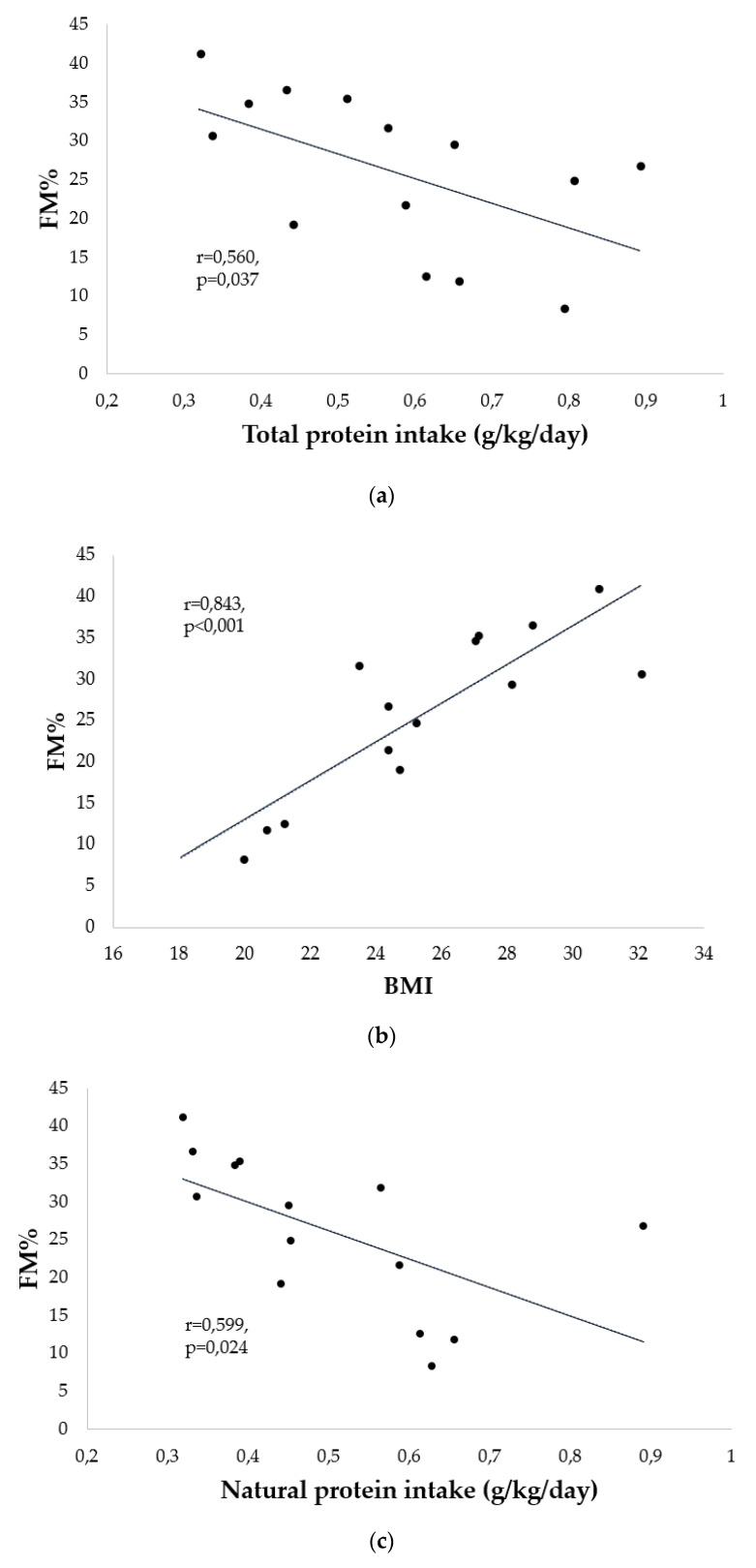
Correlations between body composition and protein intake and BMI: (**a**) FM% and total protein intake; (**b**) FM% and BMI; (**c**) FM% and natural protein intake.

**Table 1 nutrients-14-00467-t001:** Patients’ general characteristics: sex (55.5% men 44.4% women); age (years); disease (OMIM; 33.3% organic acidemias, 66.7% urea cycle disorders); body mass index (kg/m^2^); energy intake (kcal/kg/day); natural protein intake (g/kg/day); protein equivalent intake (g/kg/day, only 33.3% of subjects); total protein intake (g/kg/day); physical activity level (sedentary, active or moderately active, vigorous); type of feeding (orally feeding and/or tube feeding).

Patient	Sex	Age	Disease	Body Mass Index	Energy Intake (kcal/kg/day)	Natural Protein Intake (g/kg/day)	Protein Equivalent Intake (g/kg/day)	Total Protein Intake (g/kg/day)	Physical Activity Level	Type of Feeding
1	M	19	Methylmalonic acidemia—Cobalamin C type OMIM 277400	24.4	27.3	0.89	--	0.89	Sedentary	Orally
2	F	21	Glutaric aciduria type 1 OMIM 231670	23.5	15.5	0.56	--	0.56	Sedentary	Orally
3	M	39	Ornithine transcarbamylase OMIM 300461	27.0	18.7	0.38	--	0.38	Sedentary	Orally
4	F	21	Citrullinemia OMIM 215700	28.8	18.6	0.33	0.10	0.43	Sedentary	Orally
5	M	28	Propionic acidemia OMIM 606054	25.2	28	0.45	0.35	0.80	Sedentary	Orally
6	M	34	Argininosuccinic aciduria OMIM 207900	24.4	22.6	0.59	--	0.59	Sedentary	Orally
7	F	38	Argininosuccinic aciduria OMIM 207900	30.8	16.2	0.32	--	0.32	Sedentary	Orally
8	M	36	Methylmalonic acidemia—Cobalamin B type OMIM 607568	28.1	20.9	0.45	0.20	0.65	Sedentary	Orally
9	F	26	Argininosuccinic aciduria OMIM 207900	27.1	27.9	0.39	0.12	0.51	Sedentary	Orally
10	F	26	Methylmalonic acidemia—Cobalamin B type OMIM 607568	23.9	27.4	0.74	--	0.74	Sedentary—mobilization on wheelchair	Orally
11	M	19	Argininosuccinic aciduria OMIM 207900	20.7	31.2	0.66	--	0.66	Sedentary	Orally
12	M	17	Argininosuccinic aciduria OMIM 207900	21.2	26	0.61	--	0.61	Sedentary	Orally
13	F	34	Argininosuccinic aciduria OMIM 207900	18.1	34.2	0.64	--	0.64	Sedentary	Orally
14	F	35	Arginase deficiency OMIM 207800	27.4	21.3	0.43	0.36	0.79	Sedentary	Orally
15	M	18	Argininosuccinic aciduria OMIM 207900	24.5	22.7	0.44	--	0.44	Sedentary	Orally
16	M	28	Citrullinemia OMIM 215700	32.1	21.8	0.34	--	0.34	Sedentary	Orally
17	F	39	Isovaleric acidemia OMIM 243500	21.4	21	0.81	--	0.81	Sedentary	Orally
18	M	32	Ornithine transcarbamylase OMIM 300461	20	33.7	0.62	0.16	0.79	Sedentary	Orally
Medium values		28.6		24.9	23.7	0.54	0.22 referred to subjects on AA supplementation (33.3%)	0.61		

**Table 2 nutrients-14-00467-t002:** Biochemical parameters and reference values.

Parameter	Sample Median Value ± SD	Reference Values
Albumin	41.8 ± 3.5 g/L	35–52 g/L
Total protein	72.7 ± 4.5 g/L	64–83 g/L
Transthyretin	278 ± 70.2 mg/L	200–400 mg/L
Total cholesterol	4.5 ± 1.7 mmol/L	2.00–6.19 mmol/L
HDL cholesterol	1.1 ± 0.3 mmol/L	0.3–0.8 mmol/L
Triglycerides	1.2 ± 0.6 mmol/L	<1.69 mmol/L
Glucose	4.8 ± 0.6 mmol/L	3.7–5.6 mmol/L

**Table 3 nutrients-14-00467-t003:** Plasmatic amino acid levels and reference values, in all subjects and in the two different groups with and without AA supplementation (µmol/L).

	Sample Median Value ± Standard Deviation	Patients on AA Supplementation (39%)	Patients without AA Supplementation (61%)	Reference Values
Leucine	80.8 ± 33.1	76 ± 36.5	83.2 ± 36.5	78–160
Isoleucine	66 ± 73.9	35.3 ± 15	81.3 ± 15	34–84
Valine	145.6 ± 62.8	136.8 ± 69.2	150 ± 69.2	143–352
Lysine	132.7 ± 68.6	156.2 ± 69.1	121 ± 69.1	111–248
Methionine	41.7 ±52.2	22,7 ± 6,7	51.2 ± 6.7	14–49
Threonine	108.6 ± 35.8	110.8 ± 32.7	107.4 ± 32.7	72–168
Phenylalanine	45.6 ± 9.9	41 ± 12.3	47.9 ± 12.3	39–74

## Data Availability

Data available on request due to privacy restrictions. The data presented in this study are available on request from the corresponding author.

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
