# Peer review of "Anthropometrics, Dietary Intake and Body Composition in Urea Cycle Disorders and Branched Chain Organic Acidemias: A Case Study of 18 Adults on Low-Protein Diets"

_nutrients, 2022, doi:10.3390/nu14030467_

Round 1

Reviewer 1 Report

This paper is flawed in its design resulting in results and conclusions that are not accurately described nor assessed.  The paper was not focused on one specific topic which should have been anthropometrics.  To that effect, the title is not accurate because data collected were not complete in regards to measuring and assessing "nutritional status".  For example, the list of bioanalytes are not complete in measuring nutrition status.  The authors could have referenced currently published guidelines for management of the conditions as a benchmark for assessing the correct bioanalytes.  

Line 47/48: description on the amount of amino acid-based formulas is not well described per condition.  Patients with UCD compared to organic acidemias and tyrosinemia are quite diverse.  This case clearly be seen in the Table noting low natural protein and high formula protein contributions in one patient with tyrosinemia. 

The authors reference list seems very Eurocentric and they miss a number of references, including published guidelines for management in UCD, MMA/PA that either support or find different conclusions.   Overall, the reference list is sparse and not complete. 

Line 51:  Authors mention "Most studies..." yet only reference one study!

Line 62:  Authors note "Worldwide implementation of newborn screening, however the reference is only relevant to Europe.

Line 68-69:  Comment

Systematic Review & Meta Analysis by Rodrigues et al (with Rocha, J) showed BMI was similar between PKU and healthy controls, concluding there was "no evidence to support the idea that a Phe-restricted diet is a risk factor for the development of overweight"
Link to paper: https://doi.org/10.3390/nu13103443 

There are at least 3 other, newer publications on PKU and overweight/obesity than the 3 that the authors cite here.

Line 82: No exclusion criteria.  This to me is a major flaw which resulted in a lack of focus on this paper.  It is difficult to make conclusions on just 18 very diverse patient populations, especially understanding pathophysiology and management approaches.  In my view, the tube fed patient shouldn't have been included in this paper.  Another point I will make here and later is that there is no mention on level of ambulation among these patients, many of whom would mostly have some neurologic signs and symptoms. 

Line 84: comment 

How was this done when it a retrospective study.  Did authors exclude patients where diet data were not available?  Also, 7 days of intake only do not accurately reflect blood levels and weight.  Were subjects trained on keeping diet records?

Line 99: blood collection: Only once? Why weren't there multiple data points per subject?

Line 115:  0-7 years of age to start nutrition management is a large age difference to include.  Authors would be benefited from just focusing on those identified and treated through NBS or those not picked up by NBS.  I would think patient status would vary considerably between those started at 1 month vs. 7 years. Not a factor in metabolic control or childhood growth?

Line 121: low protein foods can provide many of the calories in a diet.  Authors should note how many total calories per day came from LP foods. 
Moreover, energy intake is notoriously underreported which can explain the lack of relationship between weight and intake.
Also, how can authors make any conclusions on weight which is a long term effect and compare to 7 days of intake.

Line 127:  natural protein intake on pts taking AA supplements was 0.61 whereas pts on only natural protein was 0.7.  this is not clinically different especially in a cohort of 18 pts.  Pts on aa supplements had total protein closer to WHO safe levels.  If that is the case then wouldn't the conclusion be that AA supplements should be used?

Line 130:  If AA based formulas only provided 9% of the total energy then how can the authors mention, without explaining, why pts on aa formulas weighed more.  The authors are intimating that aa formulas result in higher weight but it is only an observation and honestly cannot be stated with any consideration on reproducibility.

Line 134:  no statistical nor clinical difference.

Line 141:  Low leucine and other essential AA may be showing poor total and natural protein intake.  Moreover, if long term can influence FFM vs FM.  Authors do not mention this in paper.  Also, only 30% of pts met BCAA requirements.

Line 141: What is meant by "lower than expected"? Lower than requirements?

Line 145: Intakes of CHO and FAT are within recommended ranges.  I assume these include those with those on AA supplements.  Authors should not any difference between those only on NP and those on NP and AA supplements.   Based on the overall low protein intakes wouldn't the conclusion be that intakes of non-protein energy sources are the result of weight issues and not amino acid-based formulas, especially considering only 39% subjects were consuming these formulas.

Line 156:  Compare with normal adult European population?

Figure 1: How does this correlate to plasma levels? That might be a more helpful figure - showing both.

Figure 2: I find these graphics don't give much context and would benefit from a more robust explanation here. Also, p-values? Are any of these differences statistically significant?  These should be noted in the figure.

Figure 3: Need information on FM and energy intake.  Cannot look at natural protein alone.  Also, what is the difference between groups.  Authors should have separated and studied the groups on formula vs no formula.

Table 1: If authors want to make conclusions about extra calories provided by formula, need to give more data on the energy intake from food and energy intake from formula and show if there's a significant difference in energy intake between those on formula and those not on formula.

Natural protein intake for most of these subjects is too low, even for UCD patients who need to restrict all protein.  Authors fail to mention this fact and give reference to published guidelines.  From the data it appears that authors themselves are not following the published european guidelines!  

Subject 13: This patient is underweight, which is not discussed. It's odd to me to lump this patient in with the other pts with BMI <25.

Table 1 Figure "TOTAL" information:  the values in this row need clarification. For most, these are averages (not totals). For Protein Equivalent, value is the average only of the 7 taking formula (not the average for all 18 subjects).

Table 2:  Authors discuss the values "recommended for disease risk reduction". Are these reference values in line with those?

Line 204: Albumin and prealbumin are widely viewed as poor indicators of nutritional status

Lines 216-220:  This is incongruous with the author's report on pg. 3 that AA supplements only made up 9% of daily kcal intake.  Authors fail to mention effect of low protein foods!  Moreover, not all amino acid-based formula contain fat and carbohydrate.  There is no mention on the type of formulas consumed. 

Lines 218-219:  need more of an explanation here - further discussion of Minoli [16] findings to make these statements

Line 237:  What are the values for risk prevention? Citations? Should list those values in Table 2 along with reference values.

Line 242:  If sharing that your data supports the findings from reference 5, we need to have visibility of that data -- Table 2 should have lab results split out by disorder category and by pts on formula vs. no formula

Line 248:  First mention of vitamins and minerals. Vit/Min status in patients or of patient's intakes not discussed in results.

References:  Authors rely heavily on this publication for background information. Is there another citation that's newer?  This article is from 2015 and there has to be more updated papers.  Authors did not do a sufficient job of searching for relevant publications. 

Author Response

Comments and Suggestions for Authors

This paper is flawed in its design resulting in results and conclusions that are not accurately described nor assessed.  The paper was not focused on one specific topic which should have been anthropometrics.  To that effect, the title is not accurate because data collected were not complete in regards to measuring and assessing "nutritional status".  For example, the list of bioanalytes are not complete in measuring nutrition status.  The authors could have referenced currently published guidelines for management of the conditions as a benchmark for assessing the correct bioanalytes.  

Thank you for your precious suggestions: we changed the title and revised references with current guidelines for these diseases.

Line 47/48: description on the amount of amino acid-based formulas is not well described per condition.  Patients with UCD compared to organic acidemias and tyrosinemia are quite diverse.  This case clearly be seen in the Table noting low natural protein and high formula protein contributions in one patient with tyrosinemia. 

Yes, we managed to described more in detail natural protein and amino acid formulas amount in guidelines (both UCD, BCOA and AApathies) and in our sample. 

The authors reference list seems very Eurocentric and they miss a number of references, including published guidelines for management in UCD, MMA/PA that either support or find different conclusions.   Overall, the reference list is sparse and not complete. 

We added current guidelines for Aminoacidopathies, UCD and MMA/PA, with a precise list.

Line 51:  Authors mention "Most studies..." yet only reference one study!

Thank you, we added more references.and modified the sentence. 

Line 62:  Authors note "Worldwide implementation of newborn screening, however the reference is only relevant to Europe.

We specified in Europe, than you for your comment. 

Line 68-69:  Comment

Systematic Review & Meta Analysis by Rodrigues et al (with Rocha, J) showed BMI was similar between PKU and healthy controls, concluding there was "no evidence to support the idea that a Phe-restricted diet is a risk factor for the development of overweight"

Link to paper: https://doi.org/10.3390/nu13103443 

There are at least 3 other, newer publications on PKU and overweight/obesity than the 3 that the authors cite here.

Thank you for your advice, we added this interesting reference. This paper (paediatric population), however, showed significant increased BMI in classic PKU patients subgroup. This finding supports our data regarding and increased BMI associated with lower natural protein intake and higher AA supplementation intakes.

Line 82: No exclusion criteria.  This to me is a major flaw which resulted in a lack of focus on this paper.  It is difficult to make conclusions on just 18 very diverse patient populations, especially understanding pathophysiology and management approaches.  In my view, the tube fed patient shouldn't have been included in this paper.  Another point I will make here and later is that there is no mention on level of ambulation among these patients, many of whom would mostly have some neurologic signs and symptoms. 

Thank you for your comment. We included the tube fed patient because he is not on total enteral feeding: he only assumes AA supplements and water by PEG and all other foods by mouth. We specified in the text and in Table 1 about this patient. 

We also added the level of ambulation of patients in the results and in Table 1.

We decided to include only IEIPM patients on LPD in adult age, that was our inclusion criteria. 

Line 84: comment 

How was this done when it a retrospective study.  Did authors exclude patients where diet data were not available?  Also, 7 days of intake only do not accurately reflect blood levels and weight.  Were subjects trained on keeping diet records?

7 days food record is routinely assessed in our patients, since they follow a special diet therapy (we trained them on keeping records). We did not exclude patients because dietaty data were always available. 

Line 99: blood collection: Only once? Why weren't there multiple data points per subject?

Yes, we have multiple data points of our patients for dietary intake, anthropometrics and blood tests because our patients are routinely assessed (visits every 6-12 months). Since our patients were metabolic stable, blood samples and dietary intake were quite representative of their clinical course.

To make more considerations on long term effects in this new population, we plan to repeat this nutritional assessment over time.

Line 115:  0-7 years of age to start nutrition management is a large age difference to include.  Authors would be benefited from just focusing on those identified and treated through NBS or those not picked up by NBS.  I would think patient status would vary considerably between those started at 1 month vs. 7 years. Not a factor in metabolic control or childhood growth?

Thank you, we described our sample more in detail in Subjects characteristics.

Line 121: low protein foods can provide many of the calories in a diet.  Authors should note how many total calories per day came from LP foods. 

Moreover, energy intake is notoriously underreported which can explain the lack of relationship between weight and intake.

Also, how can authors make any conclusions on weight which is a long term effect and compare it to 7 days of intake.

A potential underreported or not precise food intake is an intrinsic risk of every nutritional research method.  In the nutritional research field, the 7 days record diary is regarded to be the most accurate in estimating nutrient and energy intakes.

We did not calculate the energy intake by LP foods because we focused on natural and AA supplementation intake. Also, energy intake was below energy requirements in all patients, both with or without use of LP foods. 

Line 127:  natural protein intake on pts taking AA supplements was 0.61 whereas pts on only natural protein was 0.7.  this is not clinically different especially in a cohort of 18 pts.  Pts on aa supplements had total protein closer to WHO safe levels.  If that is the case then wouldn't the conclusion be that AA supplements should be used?

Thank you for your comment. Before our study, we also totally agreed with your conclusion. Dividing the cohort of patients in two subgroups - with or without AA supplementation, we noted worse nutritional outcomes in BMI, %FM, plasmatic AA  levels (in particular BCAA) in the AA supplementation group.  Therefore we wondered if there was a better practice in protein prescription, considering not only that total protein should be close to requirements but also the quality source (natural food vs AA supplements).

Line 130:  If AA based formulas only provided 9% of the total energy then how can the authors mention, without explaining, why pts on aa formulas weighed more.  The authors are intimating that aa formulas result in higher weight but it is only an observation and honestly cannot be stated with any consideration on reproducibility.

Thank you, we agree with your comment. Our findings are unexpected and more prospective data about this special population are needed to get more significant results.  As a possible explanation, we hypothesized that, together with a different protein source, in this group also a low natural protein intake can influence a lower Free Fat Mass and a higher BMI.

Line 134:  no statistical nor clinical difference.

Thank to your suggestion. We added a significance p value  for the difference in medium BMI of the two subgroups. 

Line 141:  Low leucine and other essential AA may be showing poor total and natural protein intake.  Moreover, if long term can influence FFM vs FM.  Authors do not mention this in paper.  Also, only 30% of pts met BCAA requirements.

Thank you for your advice, we added more details on AA intake and AA plasmatic levels both in the results and in the discussion, focusing on BCAA.

Line 141: What is meant by "lower than expected"? Lower than requirements?

Yes, we corrected as you advised, thank you. 

Line 145: Intakes of CHO and FAT are within recommended ranges.  I assume these include those with those on AA supplements.  Authors should not any difference between those only on NP and those on NP and AA supplements.   Based on the overall low protein intakes wouldn't the conclusion be that intakes of non-protein energy sources are the result of weight issues and not amino acid-based formulas, especially considering only 39% subjects were consuming these formulas.

Thank you for your comment. The two subgroups did not present a significant difference in energy intake  TDEE (T Test with p-value p=0.08). We specified this on paper. 

Line 156:  Compare with normal adult European population?

We specified on paper we compared our datas with Italian Adult Popolation Reference Values (ref 20).

Figure 1: How does this correlate to plasma levels? That might be a more helpful figure - showing both.

Yes, thank you. We added more figures and tables about it.

Figure 2: I find these graphics don't give much context and would benefit from a more robust explanation here. Also, p-values? Are any of these differences statistically significant?  These should be noted in the figure.

We implemented the explanation of the figure, thank you for your comment. We could not perform a T Test because it is not possible to compare a set of data with a reference range. It would be possible to perform a T Test if we had a control group, but we unfortunately we don’t have it.

Figure 3: Need information on FM and energy intake.  Cannot look at natural protein alone.  Also, what is the difference between groups.  Authors should have separated and studied the groups on formula vs no formula.

FM did not significantly correlate with energy intake in our sample. Moreover, TDEE was not significantly different in the two subgroups, so we preferred to analyze a relationship in the total sample between protein intake and FM%.

Table 1: If authors want to make conclusions about extra calories provided by formula, need to give more data on the energy intake from food and energy intake from formula and show if there's a significant difference in energy intake between those on formula and those not on formula.

As already mentioned, there was not a significant difference in energy intake between those on formula and those not on formula.

Natural protein intake for most of these subjects is too low, even for UCD patients who need to restrict all protein.  Authors fail to mention this fact and give reference to published guidelines.  From the data it appears that authors themselves are not following the published European guidelines!  

Thank you for your comment, it is a controversial finding and we discussed it better on paper. We do follow the European Guidelines, and our patient’s protein tolerance is individualized as suggested on the guidelines themselves, resulting below WHO recommendations.   

Subject 13: This patient is underweight, which is not discussed. It's odd to me to lump this patient in with the other pts with BMI <25.

We added more details in “Subjects characteristics”.

Table 1 Figure "TOTAL" information:  the values in this row need clarification. For most, these are averages (not totals). For Protein Equivalent, value is the average only of the 7 taking formula (not the average for all 18 subjects).

Thank you, last row description was confusing indeed. We fixed that.  

Table 2:  Authors discuss the values "recommended for disease risk reduction". Are these reference values in line with those?

Thank you, we agree our discussion was not consistent on that point, so we decided to modify the discussion.

Line 204: Albumin and prealbumin are widely viewed as poor indicators of nutritional status

Even if there would be better indicators, in our hospital albumin and prealbumin are routinely assessed as an indicator of metabolic protein balance.

Lines 216-220:  This is incongruous with the author's report on pg. 3 that AA supplements only made up 9% of daily kcal intake.  Authors fail to mention effect of low protein foods!  Moreover, not all amino acid-based formula contain fat and carbohydrate.  There is no mention on the type of formulas consumed. 

Thank you for your comment: we corrected the sentence and specified this better on discussion. It is true not all formulas contain fats and maltodextrins. In our sample, however, all patients consumed AA supplementation containing extra micronutrients and maltodextrins. 

Lines 218-219:  need more of an explanation here - further discussion of Minoli [16] findings to make these statements

Thank you, we added more explanation about it.

Line 237:  What are the values for risk prevention? Citations? Should list those values in Table 2 along with reference values.

As previously mentioned, we agree our discussion was not consistent on that point, so modified this point.

Line 242:  If sharing that your data supports the findings from reference 5, we need to have visibility of that data -- Table 2 should have lab results split out by disorder category and by pts on formula vs. no formula.

Thank you, we added a supplementary Table about AA plasmatic levels in the two different subgroups.

Line 248:  First mention of vitamins and minerals. Vit/Min status in patients or of patient's intakes not discussed in results.

Thank you for your comment, we added some information about vitamins and minerals intake and supplements in the results.

References:  Authors rely heavily on this publication for background information. Is there another citation that's newer?  This article is from 2015 and there has to be more updated papers.  Authors did not do a sufficient job of searching for relevant publications. 

Thank you for your suggestion, we added some new relevant references.

Reviewer 2 Report

Review of the retrospective study evaluating nutritional status of IEIPM adult patients consuming Low Protein Diets. Manuscript has a potential of novelty and originality. Interest to the readers and soundness are fair. However, the quality of presentation should be improved. It seems that this is a pending version of the draft not fully completed. Some aspects are enlisted below:

  1. Contradictory findings are found in abstract “Abstract line 29-31 and Results 163-165.
  2. 3.1.2. Dietary intake – description of the diet is unclear and it should be more precisely described
  3. Measurement of body composition by impedance should be described better and clearer.
  4. Patient general characteristics table should be completed by type of lifestyle, glucose levels, insulin level and free fatty acids levels

5. Descriptions of tables and figures should be expanded and improved. These descriptions are not completed.

Author Response

Open Review

Comments and Suggestions for Authors

Review of the retrospective study evaluating nutritional status of IEIPM adult patients consuming Low Protein Diets. Manuscript has a potential of novelty and originality. Interest to the readers and soundness are fair. However, the quality of presentation should be improved. It seems that this is a pending version of the draft not fully completed. Some aspects are enlisted below:

 Thank you for your comments. We checked the English form and we improved the quality of the presentation, in order to be more clear in our intent to describe nutritional aspects of this new adult IEIPM population.

  1. Contradictory findings are found in abstract “Abstract line 29-31 and Results 163-165.

We corrected it in the results section, thank you.

  1. 3.1.2. Dietary intake – description of the diet is unclear and it should be more precisely described

Thank you, we detailed dietary intake by dividing patients into those taking AA supplements and those not taking AA supplements. We added some data about AA intake. 

  1. Measurement of body composition by impedance should be described better and clearer.

Thank you for your comment. We added some details in the section Materials and Methods. 

  1. Patient general characteristics table should be completed by type of lifestyle, glucose levels, insulin level and free fatty acids levels.

Thank you for your comment, we added in the table the type of lifestyle. Unfortunately, insulin level was not routinely collected for our patients.

  1. Descriptions of tables and figures should be expanded and improved. These descriptions are not completed.

We added more figures and tables about nutritional intakes and nutritional status, with more descriptions. We also divided our sample in two subgroups (with or withour AA supplements) to better describe our findings. Thank you.

Round 2

Reviewer 1 Report

I thank the authors for their efforts in improving this submission.  Overall, I strongly believe this paper undergo a major revision with exclusion of subjects 15 and 18, both of whom have aminoacidopathies.  This group of inborn errors is significantly different than urea cycle disorders and organic acidemias in the fact that for subjects 15 and 18 use of AABF constitutes a significantly higher percentage of nutrition management compared to UCD and OA, where a low natural protein diet ALONE, without AABF is an option for management.  Because the authors added these two patients the data on how much AABF is skewed.  If you look at Table 1 there are only 7 of 18 patients (39%) consuming AABF.  Total mean protein intake of all 7 was 0.38 g/kg/day.  However, if you take out patients 15 and 18 the mean drops significantly to only 0.23g/kg/day: for patients 15 and 18 the average is 0.765g/kg/day.  Consequently, these two patients significantly skew the data, and therefore Results and Conclusions.  To me, this is a major flaw in the paper.

On another point, the authors used sensational language like "alarming" when describing "prevalence" of obesity and overweight.  This language is not only sensationalism but their results based on 18 case studies disallows such use of this word, especially when only 9% energy is coming from amino acid-based formulas AABF. 

Authors still do not know add sufficient data on amount of carbohydrate and fat from AABF and total diet.  Without these data, and focus on protein, how can the authors make comparisons on outcomes based solely on AABF without going into more detail the composition of these formulas. 

Section 3.1.2:  Overall, almost all these patients are on too low of both total and natural protein based on references and WHO recommendations.   Authors do not discuss in conclusions the need to assure patients get adequate protein in their diet and how best to prescribe AABF.  Moreover, if total and natural protein are low then authors need to go into more discussion on amounts of carbohydrate and fat consumed, both of which in excess result in overweight and contribute to obesity.  

What is confusing is that only only 9% of the total calories are from AABFs? So if these formulas only provide 9% of total energy intake then how can the authors make any strong conclusion on differences in body composition?  Moreover, when you look at the total dietary energy intake between AABF intake and non-AABF group there is no significant difference.  Consequently, how can the authors make any clinical connection between differences in body mass index, "prevalence" to obesity and overweight? 

In Conclusion authors note that both low total and natural protein intake were related to fat mass %.  However, the data show that only total protein was significantly, but weakly related to fat mass. 

In  Conclusion authors state that " Dividing the cohort in patients in two subgroups - with or without AA supplementation, we noted worse nutritional outcomes in BMI, %FM, plasmatic AA levels (in particular BCAA) in the AA supplementation group, without significant differences in energy intake".  Use of the word "worse" is not only inaccurate but subjective.   When I look at the plasma concentrations of amino acids only valine and leucine are below reference ranges.  However, reference ranges are quite variable and dependent on a number of confounders.  Importantly, are the valine and leucine levels in ranges of clinical significance?  Having plasma concentrations below reference range, especially when protein analytes were normal, dictate a deficiency.  In place of the word "worse" use terms like "lower than"...

Conclusions, paragraph 7, line 1 authors discuss MMA/PA but reference #16 is in patients with PKU.  Authors should note this or get a better reference.

Figure 3: not sure what the colors define.  Recommendation is to group both b and c into one bar graph for clearer representation of data.

In Conclusion authors note "Vitamin and mineral supplements specific for LPD are needed [2-3,6-9], but most of these supplements are born for paediatric age and not well tailored for adulthood".  From my knowledge there are no specific low protein diet vitamin and mineral supplements neither in pediatrics nor adults.  Clinicians monitor status and then prescribe over-the-counter, or if needed, medicinal prescriptions to either maintain normal nutrition status or correct a deficiency.  My point is that there many over the counter vitamin and mineral supplements for adults, but none specific for inborn errors.  

I do agree with the authors and referenced citations that physical activity should be an important part of management of inborn errors.  However, this is a challenge with patients who are not ambulatory.  

Author Response

Open Review

I thank the authors for their efforts in improving this submission.  Overall, I strongly believe this paper undergo a major revision with exclusion of subjects 15 and 18, both of whom have aminoacidopathies.  This group of inborn errors is significantly different than urea cycle disorders and organic acidemias in the fact that for subjects 15 and 18 use of AABF constitutes a significantly higher percentage of nutrition management compared to UCD and OA, where a low natural protein diet ALONE, without AABF is an option for management.  Because the authors added these two patients the data on how much AABF is skewed.  If you look at Table 1 there are only 7 of 18 patients (39%) consuming AABF.  Total mean protein intake of all 7 was 0.38 g/kg/day.  However, if you take out patients 15 and 18 the mean drops significantly to only 0.23g/kg/day: for patients 15 and 18 the average is 0.765g/kg/day.  Consequently, these two patients significantly skew the data, and therefore Results and Conclusions.  To me, this is a major flaw in the paper.

Thank you for your suggestions. We excluded the two patients with AApathies. In this period however we observed two other patients with UCD which were not included in the previous group. Hence we added them to the sixteen patients with UCD and OA recalculating the data reported in our manuscript and their correlations. 

On another point, the authors used sensational language like "alarming" when describing "prevalence" of obesity and overweight.  This language is not only sensationalism but their results based on 18 case studies disallows such use of this word, especially when only 9% energy is coming from amino acid-based formulas AABF. 

We modified the language changing the term “alarming”. 

Authors still do not know add sufficient data on amount of carbohydrate and fat from AABF and total diet.  Without these data, and focus on protein, how can the authors make comparisons on outcomes based solely on AABF without going into more detail the composition of these formulas. 

We added the different intakes in macronutrients in the two groups and we specified the amount of carbohydrates from AABF. 

Section 3.1.2:  Overall, almost all these patients are on too low of both total and natural protein based on references and WHO recommendations.   Authors do not discuss in conclusions the need to assure patients get adequate protein in their diet and how best to prescribe AABF.  Moreover, if total and natural protein are low then authors need to go into more discussion on amounts of carbohydrate and fat consumed, both of which in excess result in overweight and contribute to obesity.  

In the discussion we added a sentence to indicate this concept.

What is confusing is that only only 9% of the total calories are from AABFs? So if these formulas only provide 9% of total energy intake then how can the authors make any strong conclusion on differences in body composition?  Moreover, when you look at the total dietary energy intake between AABF intake and non-AABF group there is no significant difference.  Consequently, how can the authors make any clinical connection between differences in body mass index, "prevalence" to obesity and overweight? 

Thank you for your comment, we added a new sentence in the discussion about teh protein leverage (overconsumption of fats and carbohydrates in response to reduction of protein intake).

In Conclusion authors note that both low total and natural protein intake were related to fat mass %.  However, the data show that only total protein was significantly, but weakly related to fat mass. 

By excluding the two AApaties patients and by adding the two UCD patients recently observed, we noted significant results in protein intake (both total and natural) related to %FM.

In  Conclusion authors state that " Dividing the cohort in patients in two subgroups - with or without AA supplementation, we noted worse nutritional outcomes in BMI, %FM, plasmatic AA levels (in particular BCAA) in the AA supplementation group, without significant differences in energy intake".  Use of the word "worse" is not only inaccurate but subjective.   When I look at the plasma concentrations of amino acids only valine and leucine are below reference ranges.  However, reference ranges are quite variable and dependent on a number of confounders.  Importantly, are the valine and leucine levels in ranges of clinical significance?  Having plasma concentrations below reference range, especially when protein analytes were normal, dictate a deficiency.  In place of the word "worse" use terms like "lower than"...

Thank you for your suggestion. We modified the use of the term “worse”. 

Conclusions, paragraph 7, line 1 authors discuss MMA/PA but reference #16 is in patients with PKU.  Authors should note this or get a better reference.

Thank you, we updated the reference that you cited. 

Figure 3: not sure what the colors define.  Recommendation is to group both b and c into one bar graph for clearer representation of data.

We added some information about the colours of Figure 3.

In Conclusion authors note "Vitamin and mineral supplements specific for LPD are needed [2-3,6-9], but most of these supplements are born for paediatric age and not well tailored for adulthood".  From my knowledge there are no specific low protein diet vitamin and mineral supplements neither in pediatrics nor adults.  Clinicians monitor status and then prescribe over-the-counter, or if needed, medicinal prescriptions to either maintain normal nutrition status or correct a deficiency.  My point is that there many over the counter vitamin and mineral supplements for adults, but none specific for inborn errors.  

I do agree with the authors and referenced citations that physical activity should be an important part of management of inborn errors.  However, this is a challenge with patients who are not ambulatory. 

We agree with your opinion about vitamin and mineral supplements and about the importance of physical activity, thank you. 

Reviewer 2 Report

Now the manuscript is written clearer, however, major comment to the manuscript now is:

Why did not you display differences in macronutrients (carbohydrate and fat) intake between patients consuming natural protein diet and that on AA supplementation. In other words you should collect more detailed information about diet composition (carbohydrates vs fat) and metabolism (insulin, glucose levels) to understand why patients in AA supplementation group become obese. Perhaps simple cbh content is less favourable in AA group e.g. relatively big amounts of maltodextrins. More attention should be paid to general metabolism rather than concentrate solely on proteins.

Minor comment: die is not day in English.

Author Response

Open Review

Comments and Suggestions for Authors

Now the manuscript is written clearer, however, major comment to the manuscript now is:

Why did not you display differences in macronutrients (carbohydrate and fat) intake between patients consuming natural protein diet and that on AA supplementation. In other words you should collect more detailed information about diet composition (carbohydrates vs fat) and metabolism (insulin, glucose levels) to understand why patients in AA supplementation group become obese. Perhaps simple cbh content is less favourable in AA group e.g. relatively big amounts of maltodextrins. More attention should be paid to general metabolism rather than concentrate solely on proteins.

Thank you for your suggestions. We added some information about diet composition of the two different groups. 

Minor comment: die is not day in English.

Thank you, we correct with the right form “day” both in the text and the figures.